# Effective Components and Molecular Mechanism of Agarwood Essential Oil Inhalation and the Sedative and Hypnotic Effects Based on GC-MS-Qtof and Molecular Docking

**DOI:** 10.3390/molecules27113483

**Published:** 2022-05-28

**Authors:** Canhong Wang, Yunyun Wang, Bao Gong, Yulan Wu, Xiqin Chen, Yangyang Liu, Jianhe Wei

**Affiliations:** 1Hainan Provincial Key Laboratory of Resources Conservation and Development of Southern Medicine, Hainan Branch of the Institute of Medicinal Plant Development, Chinese Academy of Medical Sciences, Peking Union Medical College, Haikou 570311, China; xinzhuangjianpo@163.com (C.W.); cloud22wang@163.com (Y.W.); gongbao0112@aliyun.com (B.G.); wyl190903008@163.com (Y.W.); 18359033886@163.com (X.C.); 2Key Laboratory of State Administration of Traditional Chinese Medicine for Agarwood Sustainable Utilization, Hainan Branch of the Institute of Medicinal Plant Development, Chinese Academy of Medical Sciences, Peking Union Medical College, Haikou 570311, China; 3Key Laboratory of Bioactive Substances and Resources Utilization of Chinese Herbal Medicine, Institute of Medicinal Plant Development, Chinese Academy of Medical Sciences, Peking Union Medical College, Beijing 100193, China; 4National Engineering Laboratory for Breeding of Endangered Medicinal Materials, Ministry of Education, Institute of Medicinal Plant Development, Chinese Academy of Medical Sciences, Peking Union Medical College, Beijing 100193, China

**Keywords:** agarwood essential oil, incense inhalation, sedative and hypnotic effects, pharmacodynamic substance, Glu–GABA balance, molecular docking

## Abstract

Agarwood has been used for the administration of hypnotic therapy. Its aromatic scent induces a relaxed state. However, its aromatic constituents and the underlying molecular effect are still unclear. This study aims to determine the active substance and molecular mechanism of the hypnotic effect of agarwood essential oil (AEO) incense inhalation in insomniac mice. Insomnia models were induced by para-chlorophenylalanine (PCPA, 300 mg/kg) in mice. The sleep-promoting effect was evaluated. Neurotransmitter levels and its receptor were detected to explore the molecular mechanism. The effective components were analyzed by GC-Q/TOF-MS of AEO. The binding mechanisms of the core compounds and core targets were verified by molecular docking. These results showed that AEO inhalation could significantly shorten sleep latency and prolong sleep time, inhibit autonomous activity and exert good sedative and sleep-promoting effects. A mechanistic study showed that AEO inhalation increased the levels of γ-aminobutyric acid (GABA_A_), the GABA_A_/glutamic acid (Glu) ratio, 5-hydroxytryptamine (5-HT) and adenosine (AD), upregulated the expression levels of GluR1, VGluT1 and 5-HT1A and downregulated 5-HT2A levels. Component analysis showed that the most abundant medicinal compounds were eremophilanes, cadinanes and eudesmanes. Moreover, the docking results showed that the core components stably bind to various receptors. The study demonstrated the bioactive constituents and mechanisms of AEO in its sedative and hypnotic effects and its multicomponent, multitarget and multipathway treatment characteristics in PCPA-induced insomniac mice. These results provide theoretical evidence for insomnia treatment and pharmaceutical product development with AEO.

## 1. Introduction

With the increasing pressure of modern society, insomnia has become the most common mental disorder [1]. More than 300 million people suffer from insomnia in China, according to a survey by the Healthy China Initiative (2019–2030). Insomnia can lower the body’s immunity, and long-term insomnia can induce a variety of physical and mental diseases, which seriously affect people’s physical and mental health and quality of life [2]. At present, sedative hypnotic drugs can improve insomnia but cannot cure the disease, and long-term medication will produce dependence [3]. The mechanisms of insomnia have not yet been fully explained. Abnormal levels of central neurotransmitters (5-HT, DA, GABA, Glu, etc.) and the regulation of receptor function are recognized as pathogenic factors at present. Interestingly, more effective and safer herbal medicine therapies have been widely used for those suffering from sleep disorders in recent years [4,5].

Agarwood is a highly valuable aromatic resinous heartwood of *Aquilaria sinensis* (Lour.) Gilg, and it is a traditional fragrant medicine used in China, Southeast Asia and the Middle East. Traditional medicinal efficacy of “nourish qi and spirit, and cure of mind and spirit” was recorded [6]. Modern pharmacological studies have shown that the extracts [7], volatile oil [8,9] and sesquiterpenes and their derivatives [10,11,12] of agarwood have sedative, hypnotic, anti-depressive and anti-neuritis effects on the central nervous system (CNS). Previous studies in our group confirmed that intraperitoneal injection with AEO has sedative and hypnotic effects through activation of the GABAergic system. Sixty-eight compounds of AEO were analyzed and identified by GC-MS, among which 51.13% were sesquiterpenes [9], small fat-soluble volatile substances that can be used for the treatment of brain diseases due to their high blood–brain barrier permeability. Agarwood has been used as incense for more than two thousand years and it has a good tranquilizing effect [13]. Volatile substances are the main components of agarwood. Studies have reported that the inhalation of agarwood aromatics can promote sleep and improve insomnia and the sleep rhythm state, and these effects are related to the regulation of neurotransmitters [14,15,16]. Agarwood’s aromatic scent, emitted from heat or smoke inhalation, has sedative and hypnotic effects. However, its less studied volatile aromatic constituents may contribute to its activity, and the underlying molecular effect is still unclear.

This study aimed to evaluate the sedative and sleep-promoting properties of AEO inhaled medications and to explore the potential mechanisms based on the Glu–GABA and serotonergic nervous systems. In addition, the molecular mechanism was verified by confirming the effective combination of active ingredients and identifying their core targets by molecular docking.

## 2. Results

### 2.1. AEO Affects Sleep

Compared with the normal group, the model group exhibited obviously prolonged sleep latency and a shortened sleep time (*p* < 0.01), suggesting that the animals exhibited symptoms of insomnia and successful model preparation. Compared with the model group, AEO significantly shortened the sleep latency of mice in a dose-dependent manner (*p* < 0.05, *p* < 0.001) (Figure 1a) and prolonged the sleep time of the mice (*p* < 0.05, *p* < 0.01, *p* < 0.001) (Figure 1b), indicating that AEO inhalation administration has a better sleep-promoting effect. In addition, the sleep-prolonging effect of an intraperitoneal injection was slightly better than that of incense inhalation at the same dose, and the effect of incense inhalation was the same as that of the positive drug.

### 2.2. AEO Affects Autonomous Activities

Compared with that in the normal group, autonomous activity was significantly enhanced (*p* < 0.05, *p* < 0.01, *p* < 0.001) in the model group mice, suggesting that the model animals developed restlessness and that their autonomous activity was enhanced. Compared with the model group, AEO significantly shortened the exercise distance (*p* < 0.05, *p* < 0.001), decreased the velocity (*p* < 0.05, *p* < 0.01, *p* < 0.001) and prolonged the rest time (*p* < 0.05) in a dose-dependent manner, indicating that the inhalation of AEO could significantly inhibit autonomous activities and had a good sedative effect. In addition, the sedative effect of incense inhalation was the same as that of intraperitoneal injection and the positive drug (Figure 2).

### 2.3. AEO Affects Neurotransmitter Levels

The levels of GABA_A_, 5-HT and AD were significantly increased in the AEO inhalation group compared with the model group (*p* < 0.05, *p* < 0.01, *p* < 0.001), and GABA_A_/Glu was upregulated, indicating that the inhalation of AEO plays a sedative and sleep-promoting role by regulating the secretion levels of central neurotransmitters and modulating the excitatory and inhibitory neurotransmitter GABA_A_/Glu secretion balance (Figure 3).

### 2.4. AEO Affects Protein Expression

Compared with the model group, AEO inhalation significantly increased the protein expression levels of 5-HT1A, decreased the protein expression levels of 5-HT2A and increased the levels of GluR1 and VGluT1. The relative protein expression diagram directly displays the experimental results, indicating that inhalation of AEO can regulate the synthesis, secretion and metabolism of the neurotransmitters 5-HT and Glu. This indirectly affects the content of 5-HT and Glu and regulates the balance of Glu/GABA, playing a sedative and sleep-promoting role in the homeostasis of the system (Figure 4).

### 2.5. Headspace Solid-Phase Microextraction Optimization

The results showed that the optimal HS-SPME conditions applied for the AEO were: 50/30 μm DVB/CAR/PDMS as a fiber coating, as shown in Appendix A, 3 min of incubation time, 15 min of extraction time and an 80 °C extraction temperature, as shown in Appendix A. In addition, for the analysis of the serum samples, optimal conditions were 65 μm PDMS/DVB as a fiber coating, 3 min of incubation time, 20 min of extraction time, an 80 °C extraction temperature, as shown in Appendix A, and 20 mg of salt addition, as shown in Appendix A.

### 2.6. Chemical Profile of AEO and Serum

Since the two administration groups showed significant therapeutic effects, AEO circulating in the blood may play an important role in its sedative efficacy. Therefore, we analyzed AEO serum samples of the injection group and inhalation group at four time points (10 min, 30 min, 1 h and 2 h) by using the same method. The base peak chromatogram of AEO is shown in Figure 5. Seventy-nine components were identified, and the relative percentages of all of the compounds are summarized in Appendix A.

In both serum sample groups, all detected peaks consisted of silicon-derived compounds (G1–G6). Compared to the blank air GC-MS profiles, the HS-SPME extract of the control serum samples contained seven compounds (B1–B7) that were unidentified. Twenty-five agarwood oil compounds (Z1–Z25) were well separated and putatively identified as blood components, as shown in Figure 6, and their chemical structures are shown in Figure 7.

### 2.7. Component–Target Molecular Docking

The lower the binding energy is, the more stable the ligand–receptor binding conformation. As a result, a binding energy ≤ −5.0 kJ/mol was used as the screening criterion. The binding atoms, binding sites and binding energy intuitively show the interaction and stability of the docking model.

The molecular docking between 25 blood components and 4 core targets showed that the core components and targets bind significantly and stably, and the binding energies were all less than −7 kJ/mol (Appendix A). Interestingly, the binding energies of the five core components (Aromadendrene oxide 2, *gamma*-Maaliene, Aristoler, Dehydrofukinone and Spathulenol) and four core targets (GABRA1, GRIA1, HTR1A and HTR2A) were all below −9 kJ/mol, indicating that these ligands and receptors could more stably and easily form stable binding conformations. The affinity of target proteins for the core components indicated that they were closely related to and were the key targets for promoting sleep in the treatment of insomnia disorders. Additionally, the core components were successfully docked with the core targets, which may be due to the formation of hydrophobic interactions, hydrogen bonds and π-stacking between them, revealing a stable docking model with a specific binding site, binding distance and binding atom. For example, the GABRA1–Spathulenol complex was stabilized at amino acid residue ALA-300 by one hydrogen bond. Spathulenol formed one hydrogen bond with LYS-42 in GRIA1. GRBRA1 and aromadendrene oxide 2 were bound at PHE-77 by one π-stacking bond. Gamma–Maaliene interacted with GRABR1 via one π-stacking bond on TYR-210. GRBRA1 and dehydrofukinone were bound at PHE-77 and TYR-210 by three π-stacking bonds. GRBRA1 and Aristoler were bound at PHE-289 by two π-stacking bonds. The GRIA1–Aristoler complex was connected to PHE-89 by one π-stacking bond. The GRIA1–gamma-Maaliene complex was stabilized at PHE-89 by one π-stacking bond. Dehydrofukinone formed two π-stacking bonds with PHE-89 in GRIA1. The core target HTR1A and the core components (Aromadendrene oxide 2, gamma-Maaliene, Aristoler, Dehydrofukinone and Spathulenol) were bound at PHE-403 by π-stacking bonds. The HTR2A–gamma-Maaliene complex was stabilized at TYR-254 by one π-stacking bond. Table 2 shows the binding energies of the core components and core targets; Figure 8 shows the docking modes of the core GABRA1, GTIA1, HTR1A and HTR2A proteins with the key components. Figure 9 shows the predicted molecular mechanism of AEO in regulating sleep.

## 3. Discussion

Agarwood, a traditional fragrant medicine, has been used for the treatment of insomnia via aromatherapy since ancient times. Medicinal chemistry has identified that sesquiterpene compounds are effective components with sedative and hypnotic functions [9]. However, the molecular mechanism by which sesquiterpene exerts its sedative and hypnotic effects has not yet been fully elucidated. This experiment simulated aromatherapy incense and AEO inhalation to explore volatile substances and the possible mechanism. The results suggested that the inhalation of AEO better promoted the sleep effect by regulating the central excitability/inhibition (E/I) Glu–GABA amino acid neurotransmitter system secretion balance, serotonin neurotransmitter activity and receptor function. In addition, preliminary studies proved that among eight main sesquiterpene types, eremophilanes, cadinanes and eudesmanes are the three main pharmacodynamic components of agarwood for sedation and hypnosis. Moreover, molecular docking also confirmed that the core compounds stably bound to the core molecular targets to mediate their effects. This study proves the rationality of ancient aromatherapy incense and explains the scientific connotation of “nourishing qi and spirit” of agarwood in ancient records.

GABA and Glu are the main neurotransmitters in the CNS [17,18]. The secretion balance of the E/I GABA–Glu neurotransmitter system plays an important role in sleep regulation. GABAergic neurons inhibit neuronal activity by blocking excitatory synaptic transmission by glutamate receptor antagonists and activating GABAergic neurons in the hippocampus [19]. GABA plays an important role in the regulation of sleep. GABA prolongs sleeping time and restores sleep in cats with insomnia [20]. Changing the content of GABA and its receptor expression affects the process of sleep [21,22]. Glutamate (Glu) is a major excitatory neurotransmitter that is widely found in the CNS and regulates learning, memory and cognitive functions [23,24]. However, excessive glutamate induces neuronal death, leading to various neurodegenerative diseases [25]. Although Glu levels are extremely high, most are intermediate metabolites, and only a few have functions as neurotransmitters. Thus, Glu content cannot be used to determine the excited or inhibited state of neural function. Glu charges GABA by the action of glutamate decarboxylase (GAD), so the GABA–Glu ratio can be used to assess the E/I balance state of CNS function [26]. AEO inhalation increased the levels of GABA and the GABA_A_/Glu ratio, upregulated the expression levels of GluR1 and VGluT1 and exerted a significant sedative and sleep-promoting effect by regulating the GABA–Glu system balance.

Serotonin (5-HT) is a neurotransmitter and it is involved in sleep–wake cycle regulation. A reduction in 5-HT levels leads to insomnia, indicating that 5-HT can promote sleep. The 5-HT1A receptor acts as a ligand binding to serotonin, and when used as a 5-HT1A receptor agonist, it increases slow-wave sleep and reduces sleep latency. In contrast, sleep duration was significantly reduced in individuals lacking the 5-HT1A receptor [27]. AEO inhalation increased the levels of 5-HT, downregulated the expression of 5-HTA1 and 5-HT2A and played a role in sleep by regulating the serotonin system.

HS-SPME is a nondestructive technique that has the advantages of reducing laborious, time-consuming manual work and sample preparation steps. We obtained the best selected condition based on the peak area and number of molecular features from the GC-Q/TOF-MS TIC chromatogram. Equilibration time is often not considered in optimization experiments [28,29], and a 3 min equilibration time was utilized to homogenize the AEO and serum samples before extraction. We finally selected the PDMS/DVB fiber to reduce the frequency of replacing the inlet septa. The extraction time for the AEO sample was set at 15 min to suppress the carryover. This “carryover” phenomenon was also mentioned by other articles [29,30]. As seen from Appendix A, the maximum total peak area and peak number can be obtained at an extraction time of 30 min. However, a 20 min extraction time was chosen as a compromise between sensitivity and short-term analysis. NaCl can affect the ionic strength of volatile compounds released from serum samples during the HS-SPME process [29,31]. The result of the TIC peak area is different from the TIC peak number, and the reason for this, as shown in Appendix A, is that a high salt concentration can reduce the peak resolution, which causes a large difference in the TIC peak number and peak area. Finally, 20 mg NaCl was added to the serum sample in further experiments.

Comparative analysis showed that the most effective compounds are eremophilanes, cadinanes and eudesmanes, which are the main sesquiterpene types isolated from agarwood [32,33]. The cadinane sesquiterpene dehydrofukinone (Z25) has an anxiolytic-like CNS effect, and its mechanism is related to regulating the GABA_A_ receptor [34], which is consistent with our component–target molecular docking results. Accordingly, cadinane sesquiterpenes may be the main medicinal ingredients exerting sedative activity. Nevertheless, studies on the active ingredients of AEO passing through the blood–brain barrier are still limited.

The sleep-promoting effect of AEO was related to the regulation of GABA–Glu and the 5-HT nervous system and its receptors, and the core compounds and core targets were stably bound through hydrophobic interactions and π-stacking bonds. In summary, the results of this study indicate that incense inhalation of AEO has significant sedative and sleep-promoting effects, and the molecular mechanism verification results confirmed that the Glu and 5-HT receptors were the key targets and stably bound to the core compounds to promote sleep. Compounds Z4, Z16, Z17 and Z21 of AEO had sedative and sleep-promoting effects. The sleep-promoting effect has the characteristics of “multiple components and multiple targets”, and the mechanism is related to the regulation of the GABA, Glu and 5-HT nervous systems. However, the specific mechanism of the sleep-promoting effect of agarwood and its active compounds needs to be further studied and explained.

## 4. Materials and Methods

### 4.1. Materials

The raw material of agarwood was 7-year-old white wood. Agarwood was produced over 18 months by the whole-tree agarwood-inducing technique [35]. The agarwood-inducing base was located in Pingding Town, Huazzhou City, Guangdong Province. The agarwood preparations were tested against the standards of the Chinese pharmacopoeia by the Agarwood Identification Center of Hainan Branch, Institute of Medicinal Plant, Chinese Academy of Medical Sciences. AEO was prepared by steam distillation in our laboratory.

### 4.2. Reagents

Sodium pentobarbital (No: 20200113) was purchased from Merck, Rahway, NJ, USA; diazepam (No: 201805) was purchased from Beina Chuanglian Biological Technology Co., Ltd., Shanghai, China; Para-chlorophenylalanine (PCPA, No: A841909) was purchased from McLean Co., Ltd, Shanghai, China. The neurotransmitter enzyme linking immunoassay kits (Glu, 5-HT, GABA_A_, DA, all with the same No: 202008) were purchased from Beijing Bosheng Jingwei Biotechnology Co., Ltd, Beijing, China. Sodium chloride (NaCl) (AR Grade) was purchased from Sinopharm Chemical Reagent Co., Ltd. (Shanghai, China).

### 4.3. Instruments

Other instruments used: Mouse autonomous activity experiment computer online detection system (Shanghai Xin Soft Information Technology Co., Ltd., Shanghai, China, model: RD-1118-co-M4); Self-made incense burner (a large box structure of 50 cm × 50 cm × 40 cm made of Plexiglass, in which a hollow cylinder structure of 20 cm × 20 cm × 20 cm is placed, and an electronic incense burner can be placed in the middle, as shown in Figure 1); Microporous plate spectrophotometer (Thermo Fisher Science Shanghai Co., Ltd., Shanghai, China, serial number: 1510-04123). GC-Q/TOF-MS analyses were performed on a 7890B gas chromatograph coupled to a 7200 quadrupole time-of-flight (Q/TOF) mass spectrometer detector (Agilent Technologies, Santa Clara, CA, USA). The GC was equipped with an MPS Robotic autosampler (CombiPAL RSI 85 autosampler) from CTC Analytics AG (Zwingen, Switzerland). Four fibers were tested in headspace solid-phase microextraction (HS-SPME) analysis, including polydimethysiloxane/divinylbenzene (PDMS/DVB; 65 μm), carboxen/polydimethysiloxane (CAR/PDMS; 85 μm), divinylbenzene/carboxen/polydimethysiloxane (DVB/CAR/PDMS; 50/30 μm) and polyacrylate (PA; 85 μm), all of which were obtained from Sigma Aldrich (Gillingham, UK).

### 4.4. Animals

SPF-class male KM mice with a body weight range of 18–22 g were purchased from Hainan Institute of Materia Medica with production license no. SCXK (Qiong) 2019-0006. The animals were kept in the SPF-class animal room of the Hainan Institute of Materia Medica at a temperature of 20–25 °C, humidity of 50–60%, light/dark cycle of 12 h/12 h, with free drinking water and food intake and adaptive feeding for 3 days, and then the experiment was conducted. The animal care and experimental protocol used in this study was approved by the Institutional Animal Care and Use Committee of Hainan Institute of Materia Medica.

### 4.5. Animal Experiment and Administration

Healthy KM mice were randomly divided into 7 groups of 8 mice each: normal group; model group; positive control diazepam group (2.5 mg/kg); AEO low, medium and high groups (2, 4, 8 μL) [9]; and the AEO intraperitoneal injection group (40 mg/kg). Except for the normal group, the insomnia models were prepared by gavaging PCPA (300 mg/kg) for 2 consecutive days. The model succeeded in showing that the mice had the physiological characteristics of irritability and increased autonomous activity. The normal group was gavaged with a corresponding volume of saline. The positive group was given the treatment by intraperitoneal injection. AEO was inhaled by heating it in an electronic aromatherapy furnace at 80 °C for 1 h/day. Aromatherapy was given for 7 consecutive days.

### 4.6. Sleep Promotion Test

On the 5th day, 1 h after administration, pentobarbital sodium (50 mg/kg) was intraperitoneally injected at 10 mL/kg. The time elapsed between the administration of pentobarbital sodium and righting reflex disappearance was recorded as the latency of sleeping time. The time elapsed between the disappearance and reappearance of the righting reflex was considered the duration of the sleeping time.

### 4.7. Autonomous Activity and Open-Field Detection

One hour after administration on the 6th day, the autonomic activity of the mice was detected by an autonomic activity computer system. Mice were placed in the autonomous activity apparatus for adaptation for 3 min, and the test was started for 10 min. During the test, the surrounding environment was kept quiet. The number of autonomous activities and open fields traversed by the mice were recorded, and the total distance traveled, average velocity and resting time were calculated.

### 4.8. Neurotransmitters Detected by Enzyme-Linked Immunoassay in Brain Tissues

One hour after the last administration, the mice were sacrificed and blood was collected. The brain was quickly collected and rinsed with normal saline. The hippocampus was stripped off and weighed. Then, 9 volumes of normal saline with precooling beforehand were added and fully homogenized in an ice bath. The supernatant was centrifuged at 3000 r/min at 4 °C for 15 min. The precipitate was used for protein extraction. The levels of 5-HT, Glu, AD and GABAA in the brain tissue homogenate supernatant were determined according to the manufacturer’s instructions.

### 4.9. Protein Expression Detected in Brain Tissue by Western Blotting

The hippocampal tissues were homogenized in a standard RIPA buffer supplemented with a cocktail of protease and phosphatase inhibitors. The homogenate was then centrifuged at 15,000× *g* at 4 °C for 10 min. The protein concentration was determined using a BCA Protein Assay Kit. According to the molecular weight of the target protein, the appropriate concentration of separation gel was selected, the sample amount was brought to 20 μL and electrophoresed, and then the proteins were transferred to a PVDF membrane, blocked with 5% skim milk powder for 1 h and the primary antibodies against β-actin (1:2000), 5-HT1A, 5-HT2A, GluR1 and VGluT1 (1:1000) were added. The membranes were incubated overnight at 4 °C, and the secondary antibody was incubated with the membrane at room temperature for 2 h. ECL chemiluminescence was performed with a gel imager. Gel-Pro Analyzer 4.0 was used for grayscale scanning and quantitative analysis, and intuitive histograms were produced.

### 4.10. Headspace Solid-Phase Microextraction

To ensure optimal sensitivity for the wide-range detection of volatile compounds, we investigated several factors that could potentially affect the extraction efficiency, including fiber type, extraction time, extraction temperature and salt addition. First, the samples were incubated in an oven for 3 min at 80 °C (without the fiber), and then the selected fiber was applied to extract the volatile components from a 20 mL vial sealed with a magnetic screw cap. After this, the fiber was inserted into the GC injector port for thermal desorption for 3 min at 250 °C. Finally, the fiber was conditioned for 10 min at a high temperature after each analysis to avoid carryover phenomena and cross-contamination between samples.

### 4.11. Gas Chromatography–Mass Spectrometry Analysis

Separation was achieved on an HP-5MS capillary column (30 m × 0.25 mm inner diameter × 0.25 μm film thickness, Agilent Technologies, Santa Clara, CA, USA). The injector and transfer line heater temperature were 250 °C, working in splitless mode for serum sample analyses and in split mode of 20:1 for oil sample analyses. Mass spectrometry was scanned in the range of *m*/*z* 20~300, with a typical scan rate of 5 spectra/s. Detection of the target compounds was accomplished through profile acquisition mode. The Agilent 7200 Q/TOF-MS used in this work had an instrument resolution specification of 12,500 FWHM (full width at half maximum) and a mass accuracy of less than 5 ppm (or ± 0.0014u) on measurement of 1.0 pg of octafluoronaphthalene at *m*/*z* 271.9867 [36].

The mouse serum samples were detected at sampling time points of 10 min, 30 min, 1 h and 2 h. The TIC (total ion current) chromatograms of the mouse serum after the administration of AEO were established by HS-SPME-GC-Q/TOF-MS. Blank injections were performed during each time point of analysis.

GC-Q/TOF-MS piloting and data acquisition were performed using Mass Hunter B.07.00 software (Agilent Technologies, Santa Clara, CA, USA), and the peak RT tolerance was set to ±0.08 min. The threshold of the signal-to-noise ratio (S/N) was set to peak height ≥1% of the largest peak. A NIST 17 library search for putative identification was performed.

### 4.12. Component–Target Molecular Docking

The molecular docking method was used to predict the binding affinity between the core components and the target proteins, which might provide a reference for further experimental verification. AutoDock Vina 1.1.2, an open-source program for performing molecular docking and virtual screening, with higher average accuracy of binding mode predictions compared to AutoDock 4, was used to conduct the docking task [37]. The SDF files of the compounds were downloaded from the PubChem database (https://pubchem.ncbi.nlm.nih.gov/, accessed on 15 October 2021) [38] and converted to MOL2 format by Open Babel 2.4.1. Three-dimensional structures of the proteins were obtained from the Protein Data Bank (PDB) database (https://www.rcsb.org/, accessed on 15 October 2021) [39]. The ligands and receptors were prepared according to the tutorial for AutoDock Vina 1.1.2. For each structure, we deleted the water molecules, added nonpolar hydrogen, calculated the Gasteiger charge and saved them in PDBQT format. The lower the Vina score is, the higher the affinity between the ligand and receptor. The conformation with the lowest affinity was used as the best docking conformation, and PLIP was used to visualize the interaction mode [40].

### 4.13. Statistical Data Analysis

The obtained experimental data were expressed as x ± s, and the comparisons between groups were processed by SPSS 7.0 statistical software. One-way ANVOA was used for variance analysis. Comparisons between groups were performed using a *t*-test, and *p* < 0.05 was considered statistically significant.

## 5. Conclusions

To the best of our knowledge, this study is the first to show that inhaled AEO incense has sedative and hypnotic effects mediated by binding to receptors and affecting the GABAergic, glutamatergic and serotonin systems. Inhaled AEO has sedative and hypnotic effects similar to those of diazepam and slightly weaker than that of AEO intraperitoneal injection. The different compounds and types of effective components may be the reasons for the differences in efficacy associated with the administration route. The main pharmacodynamic compounds are eremophilanes, cadinanes and eudesmanes. The core compounds of AEO play their roles by regulating the Glu, GABA and 5-HT systems. However, the specific molecular mechanism was only partially revealed in this study, and additional studies are necessary to explore the mechanism further. Nonetheless, AEO could serve as a potential candidate for developing a new functional product to help treat insomnia and CNS diseases. Moreover, the results provide theoretical evidence for the development and utilization of agarwood.

## Figures and Tables

**Figure 1 molecules-27-03483-f001:**
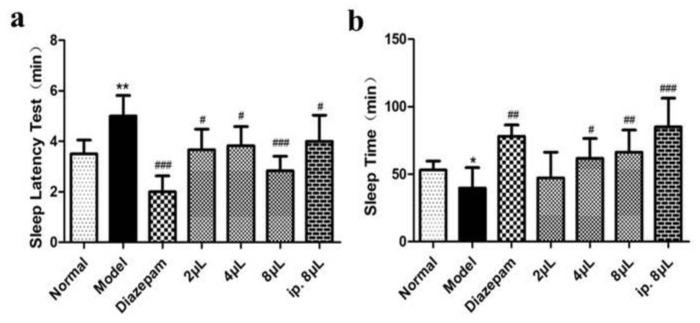
Effects on sleep of AEO inhalation. (**a**) Sleep Latency Test, (**b**) Sleep Time. All values are expressed as the means ± SD (*n* = 8). * *p* < 0.05, ** *p* < 0.01 vs. normal group; ^#^
*p* < 0.05, ^##^
*p* < 0.01, ^###^
*p* < 0.001 vs. model group.

**Figure 2 molecules-27-03483-f002:**
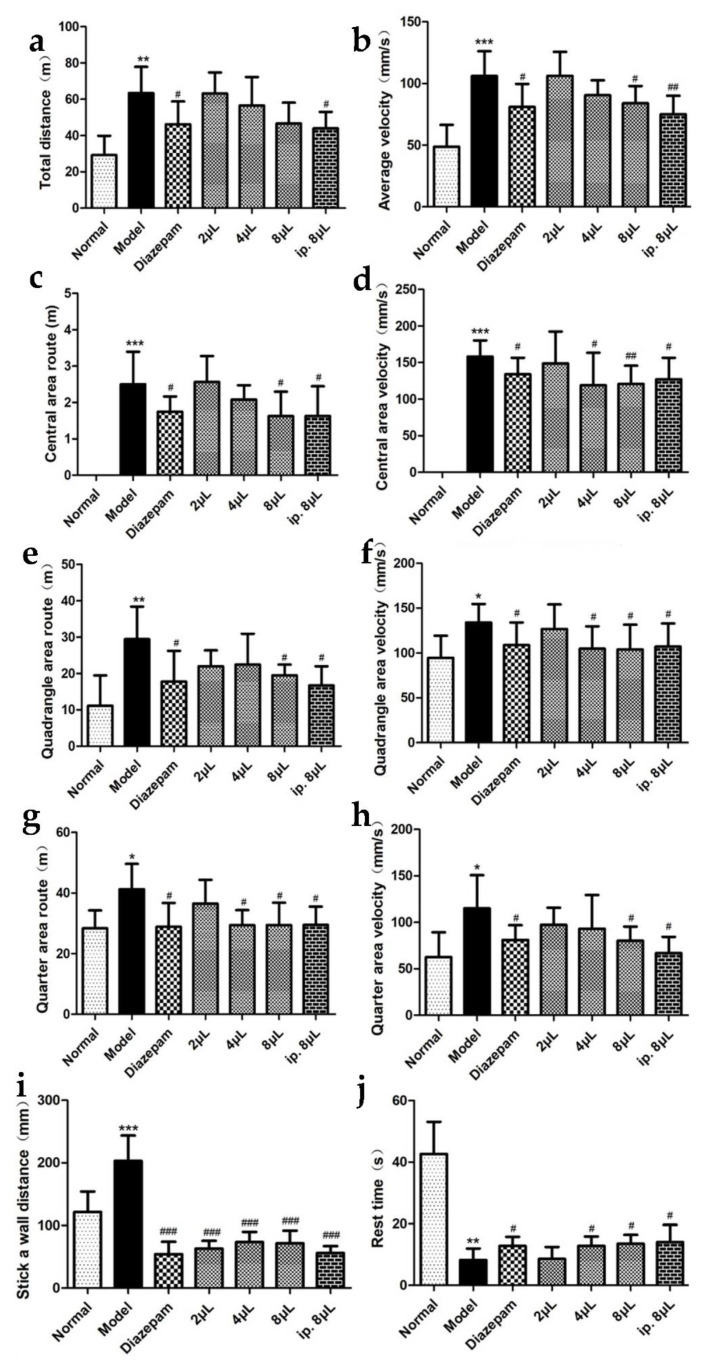
Effects on autonomous activities of AEO inhalation. (**a**) Total distance, (**b**) average velocity, (**c**) central area route, (**d**) central area velocity, (**e**) quadrangle area route, (**f**) quadrangle area velocity, (**g**) quarter area route, (**h**) quarter area velocity, (**i**) stick a wall distance, (**j**) rest time. All values are expressed as the means ± SD (*n* = 8). * *p* < 0.05, ** *p* < 0.01, *** *p* < 0.001 vs. normal group; ^#^
*p* < 0.05, ^##^
*p* < 0.01, ^###^
*p* < 0.001 vs. model group.

**Figure 3 molecules-27-03483-f003:**
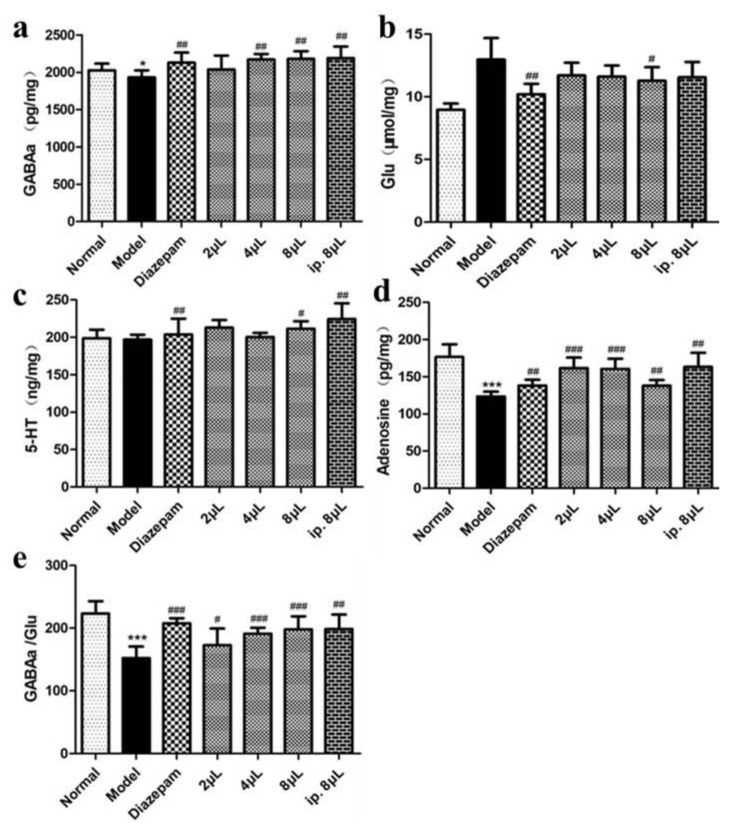
Effects on the level of neurotransmitters of AEO inhalation. (**a**) GABAa, (**b**) Glu, (**c**) 5-HT, (**d**) adenosine, (**e**) GABAa/Glu. All values are expressed as the means ± SD (*n* = 8). * *p* < 0.05, *** *p* < 0.001 vs. normal group; ^#^
*p* < 0.05, ^##^
*p* < 0.01, ^###^
*p* < 0.001 vs. model group.

**Figure 4 molecules-27-03483-f004:**
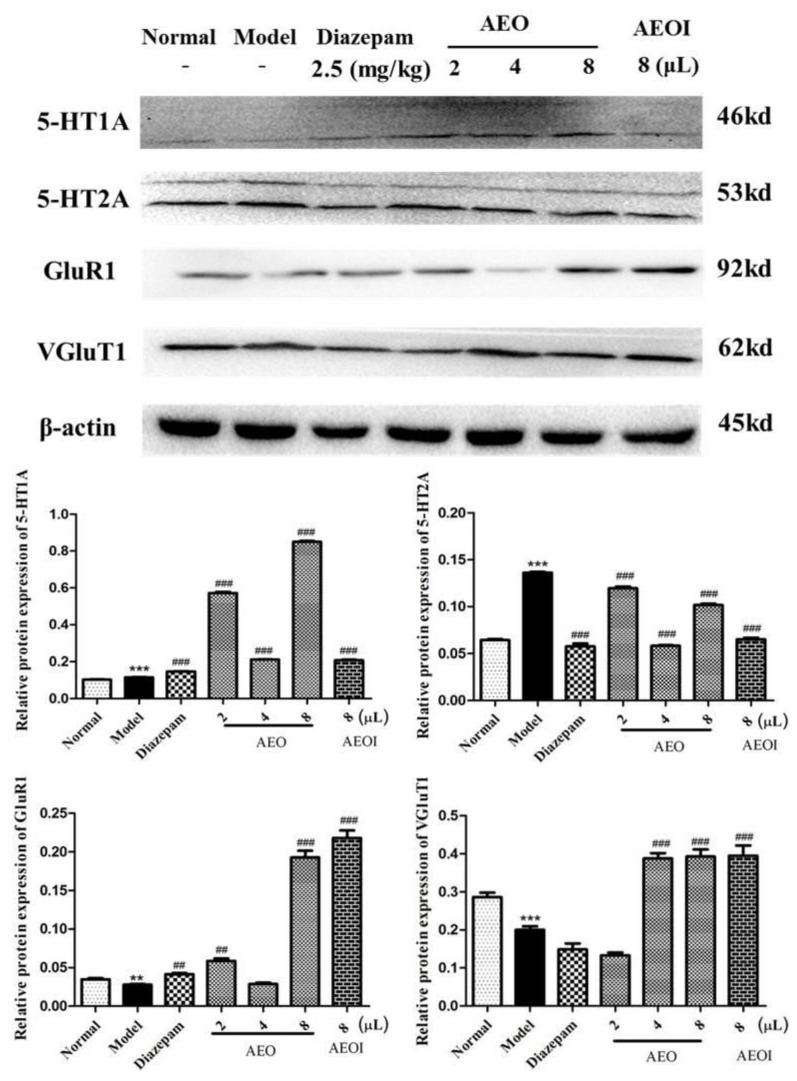
Effects on the protein levels of 5HT1A, 5HT2A, GluR1 and VGluT1 via AEO inhalation. All values are expressed as the means ± SD (*n* = 3). ** *p* < 0.01, *** *p* < 0.001 vs. normal group; ^##^
*p* < 0.01, ^###^
*p* < 0.001 vs. model group.

**Figure 5 molecules-27-03483-f005:**
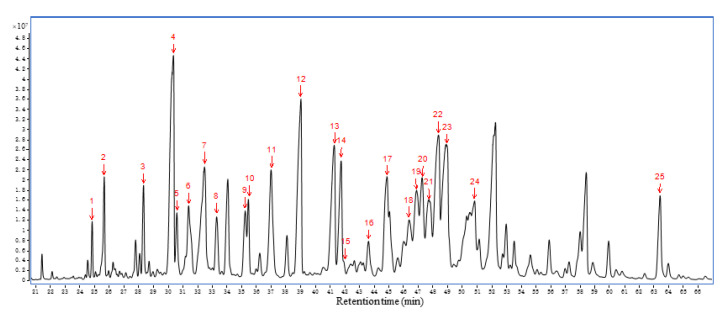
GC-MS chromatograms of HS-SPME extracts for AEO sample.

**Figure 6 molecules-27-03483-f006:**
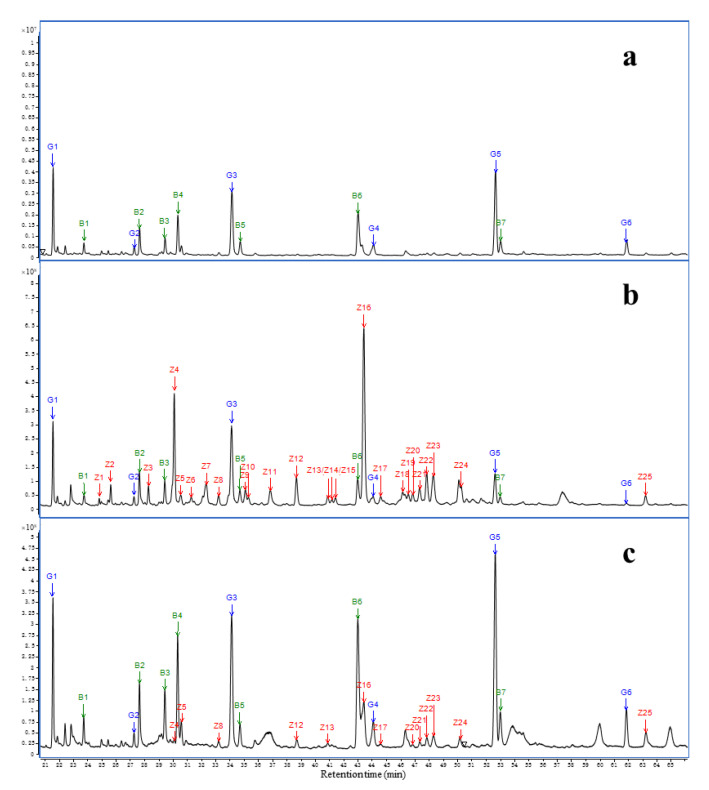
GC-MS chromatograms of HS-SPME extracts at the time points of 10 min for serum samples: (**a**) control group, (**b**) intraperitoneal injection group, (**c**) inhalation group.

**Figure 7 molecules-27-03483-f007:**
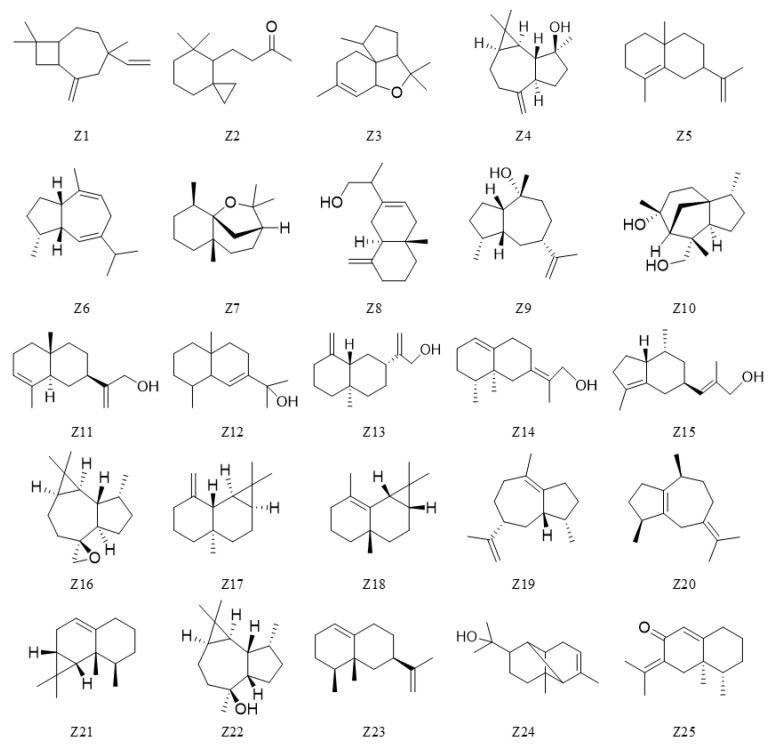
Chemical structure of 25 blood compounds identified in serum samples. From the statistical results of the total peak number and area across 4 administration time points shown in Table 1, it can be seen that the detectability of the blood components in both the intraperitoneal injection and inhalation groups reached their highest level at the time point of 10 min. We identified 25 original AEO compounds in the serum of the intraperitoneal injection group, 13 of which were identified in the inhalation group, including 24 sesquiterpenes (except Compound Z2). Compounds Z16, Z4 and Z23 were the main components in the intraperitoneal injection group (31.45%), and Compounds Z16, Z25 and Z23 were the major components in the inhalation group (4.89%). It was speculated from the blood compound results that the greater variety and greater content of the injected blood components might be the reason that the sleep-promoting effect of intraperitoneal injection was better than that of AEO inhalation.

**Figure 8 molecules-27-03483-f008:**
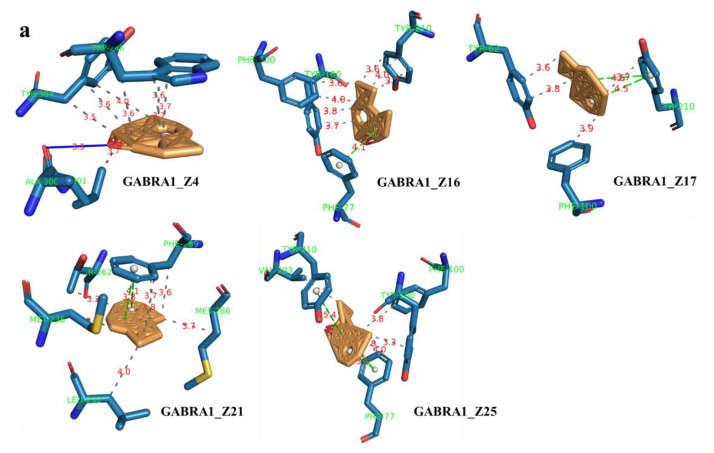
The interaction mode between five core components (Z4, 16, 17, 21, 25) and four targets. (**a**) GABRA1; (**b**) GRIA1; (**c**) HTR1A; (**d**) HTR2A; (**e**) 3D picture.

**Figure 9 molecules-27-03483-f009:**
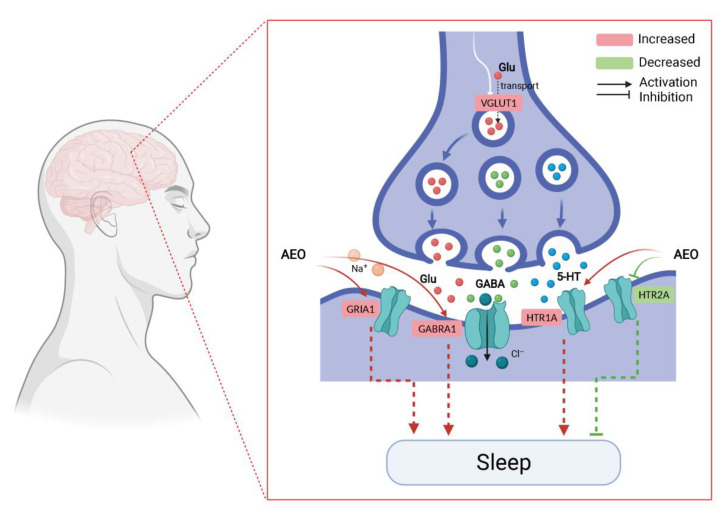
The predicted molecular mechanism of AEO-regulated 5-HT, Glu and GABA multineurotransmitter pathways in regulating sleep.

**Table 1 molecules-27-03483-t001:** Chemical composition of the 25 blood compounds identified in serum samples.

Number	RT/Min	Sesquiterpene Type	Compound	Peak Area of Intraperitoneal Injection Group/%					Peak Area of Inhalation Group/%		
				10 min	30 min	1 h	2 h	10 min	30 min	1 h	2 h
Z1	24.8	-	Bicyclo [5.2.0]nonane,2-methylene-4,8,8-trimethyl-4-vinyl-	0.31	0.34	-	-	-	-	-	-
Z2	25.6	-	Spiro [2.5]octane,5,5-dimethyl-4-(3-oxobutyl)-	1.30	0.61	0.10	-	-	-	-	-
Z3	28.3	Silphiperfolane	(1R,3aR,5aR,9aS)-1,4,4,7-Tetramethyl-1,2,3,3a,4,5a18,9-octahydrocyclopenta[c]benzofuran	1.24	0.86	0.26	-	-	-	-	-
Z4	30.1	Aromadendrane	Spathulenol	8.99	5.69	0.34	0.83	0.12	0.17	-	-
Z5	30.5	Eremophilane	4a,8-Dimethyl-2-(prop-1-en-2-yl)-1,2,3,4,4a,5,6,7-octahydronaphthalene	0.78	0.41	0.54	0.62	1.01	0.61	0.31	0.72
Z6	31.3	Guaiane	(1R,3aS,8aS)-7-Isopropyl-1,4-dimethyl-1,2,3,3a,6,8a-hexahydroazulene	0.38	0.47	-	-	-	-	-	-
Z7	32.3	-	Dihydro-beta-agarofuran	2.84	1.16	-	-	-	-	-	-
Z8	33.2	Eremophilane	(R)-2-((4aS,8aR)-4a-Methylene-1,4,4a,5,6,7,8,8a-octahydronaphthalen-2-yl) propan-1-ol	0.90	0.67	0.40	0.49	0.47	0.21	0.36	0.45
Z9	35.1	Guaiane	Pogostol	0.98	0.52	-	-	-	-	-	-
Z10	35.3	Cedrane	α-Costol	0.16	0.11	-	-	-	-	-	-
Z11	36.9	Eremophilane	2-((2R,4aR,8aR)-4a,8-Dimethyl-1,2,3,4,4a,5,6,8a-octahydronaphthalen-2-yl)prop-2-en-1-ol	2.45	1.14	-	-	-	0.26	-	-
Z12	38.7	Eremophilane	2-((4aS,8R,8aR)-4a,8-Dimethyl)-3,4,4a,5,6,7,8,8a-octahydronaphthalen-2-yl)propan-2-ol	3.02	3.82	0.53	0.32	0.69	1.01	0.29	0.28
Z13	40.9	Eremophilane	(+)-β-Costol	0.61	0.64	-	-	0.20	0.25	-	-
Z14	41.1	Cadinane	(E)-2-((8R,8aS)-8,8a-Dimethyl-3,4,6,7,8,8a-hexahydronaphthalen-2(1H)-ylidene)propan-1-ol	0.11	0.14	-	-	-	-	-	-
Z15	41.5	Brasilane	Aristol-1(10)-en-9-ol	0.49	0.35	-	-	-	-	-	-
Z16	43.5	Aromadendrane	Aromadendrene oxide-(2)	19.01	16.21	14.37	5.70	2.70	2.07	2.58	0.15
Z17	44.6	Maaliane	γ-Maaliene	0.82	0.49	-	-	0.19	0.30	-	-
Z18	46.2	Maaliane	β-Maaliene	0.31	0.42	-	-	-	-	-	-
Z19	46.6	Guaiane	Δ-Guaiene	0.44	0.45	-	-	-	-	-	-
Z20	46.8	Guaiane	β-Guaiene	0.63	0.48	-	-	0.14	0.15	-	-
Z21	47.3	Aristolane	(-)-Aristolene	1.25	1.28	-	-	0.31	0.47	-	-
Z22	47.8	Aromadendrane	Viridiflorol	2.85	3.27	0.42	-	0.61	0.75	-	-
Z23	48.3	Cadinane	Eremophilene	3.45	4.29	0.58	-	1.02	1.42	-	-
Z24	50.4	-	α-Copaen-11-ol	0.36	3.90	0.63	-	0.86	0.78	-	-
Z25	63.2	Cadinane	Dehydrofukinone	1.05	2.20	0.93	-	1.17	0.89	-	-
Relative percentage of total peak area/%				55.45	49.92	19.10	7.96	9.76	6.53	3.54	1.60
Number of total peaks				25	25	11	5	13	12	4	4

**Table 2 molecules-27-03483-t002:** Docking results of five core components with four targets.

Ligands	Compound Names	PubChem_CID	Receptors	Affinity (kcal/mol)
Z16	Aromadendrene oxide 2	16211192	GABRA1	−11.3
Z17	gamma-Maaliene	21775138	GABRA1	−11.5
Z21	Aristoler	530421	GABRA1	−10
Z25	Dehydrofukinone	177072	GABRA1	−10.8
Z4	Spathulenol	92231	GABRA1	−9.6
Z16	Aromadendrene oxide 2	16211192	GRIA1	−10.7
Z17	gamma-Maaliene	21775138	GRIA1	−10.6
Z21	Aristoler	530421	GRIA1	−10.7
Z25	Dehydrofukinone	177072	GRIA1	−10
Z4	Spathulenol	92231	GAIA1	−10.2
Z16	Aromadendrene oxide 2	16211192	HTR1A	−9.5
Z17	gamma-Maaliene	21775138	HTR1A	−9.5
Z21	Aristoler	530421	HTR1A	−9.5
Z25	Dehydrofukinone	177072	HTR1A	−9.2
Z4	Spathulenol	92231	HTR1A	−9.2
Z16	Aromadendrene oxide 2	16211192	HTR2A	−10.6
Z17	gamma-Maaliene	21775138	HTR2A	−10.3
Z21	Aristolene	530421	HTR2A	−10.3
Z25	Dehydrofukinone	177072	HTR2A	−9.6
Z4	Spathulenol	92231	HTR2A	−10.2

## Data Availability

Materials and data are available from the first author or corresponding author.

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
