# Peer review of "Effective Components and Molecular Mechanism of Agarwood Essential Oil Inhalation and the Sedative and Hypnotic Effects Based on GC-MS-Qtof and Molecular Docking"

_molecules, 2022, doi:10.3390/molecules27113483_

Round 1
Reviewer 1 Report
Dear authors,
After carefully reading your manuscript, I have the following comments and suggestions.
Please correct all citations, they must be superscript as indicated in the author guidelines.
Check scientific names are in italics.
Table after line 167 has no table caption. Also, please add the column of the mass of the metabolites.
Please improve the quality of the chromatograms in Figures 5 and 6.
Caption in figure 7. Please enrich the figure caption, by blood compounds, do you mean Agarwood plasmatic metabolites?
Please improve the quality of Figure 8
Please enrich figure 9 caption, this way your potential readers could understand better the summarized hypothesized mechanism of action.
In the Animal experiment and administration, did you use a negative control?
Please specify the protein extraction method.
Author Response
Dear reviewer,
Thanks for your the constructive and insightful comment, we have carefully studied and revised the manuscript. Please see the attachment. Thank you!

Reviewer 2 Report
the authors showed the effect of agarwood essential oil inhalation on the sedative and hypnotic, identified the major component using GC-MS as well as they suggested the mode of action of these component using molecular docking. The study looks good however some changes should be carries out
-the style of references in the manuscript should follow the journal style
-chemical components should be redrawn showing the different stereochemistry
-Figure 8 looks so crowdy, and it is better to move to supplement material and add the key structures only
-3D pics for the docking study should be included
Author Response
Dear reviewer,
Thanks for your comment. We have recised the manuscript, please see the attachment.

Round 2
Reviewer 1 Report
Dear authors,
Thank you for attending the reviewer's comments. I just have a few more suggestions to improve the format quality of the manuscript.
It seems that in some reviewed figures the old figure can still be seen, I assume this is due to the track changes tool. Please be sure figures are not repeated in the final draft of the manuscript.
Figure 8 still looks in low quality, I suggest you include higher quality images so readers can appreciate your interesting results.
I hope my comments are useful,
Good luck and kind regards,
Reviewer 2 Report
The authors did great job in the revised manuscript and look better now